# PROTOTYPE RECALLS FOR CONTINUAL LEARNING

## ABSTRACT

Continual learning is a critical ability of continually acquiring and transferring knowledge without catastrophically forgetting previously learned knowledge. However, enabling continual learning for AI remains a long-standing challenge. In this work, we propose a novel method, Prototype Recalls, that efficiently embeds and recalls previously learnt knowledge to tackle catastrophic forgetting issue. In particular, we consider continual learning in classification tasks. For each classification task, our method learns a metric space containing a set of prototypes where embedding of the samples from the same class cluster around prototypes and class-representative prototypes are separated apart. To alleviate catastrophic forgetting, our method preserves the embedding function from the samples to the previous metric space, through our proposed prototype recalls from previous tasks. Specifically, the recalling process is implemented by replaying a small number of samples from previous tasks and correspondingly matching their embedding to their nearest class-representative prototypes. Compared with recent continual learning methods, our contributions are fourfold: first, our method achieves the best memory retention capability while adapting quickly to new tasks. Second, our method uses metric learning for classification, and does not require adding in new neurons given new object classes. Third, our method is more memory efficient since only class-representative prototypes need to be recalled. Fourth, our method suggests a promising solution for few-shot continual learning. Without tampering with the performance on initial tasks, our method learns novel concepts given a few training examples of each class in new tasks.

## 1 INTRODUCTION

Continual learning, also known as lifelong learning, is the crucial ability for humans to continually acquire and transfer new knowledge across their lifespans while retaining previously learnt experiences Hassabis et al. (2017). This ability is also critical for artificial intelligence (AI) systems to interact with the real world and process continuous streams of information Thrun & Mitchell (1995). However, the continual acquisition of incrementally available data from non-stationary data distributions generally leads to catastrophic forgetting in the system McCloskey & Cohen (1989); Ratcliff (1990); French (1999). Continual learning remains a long-standing challenge for deep neural network models since these models typically learn representations from stationary batches of training data and tend to fail to retain good performances in previous tasks when data become incrementally available over tasks Kemker et al. (2018); Maltoni & Lomonaco (2019).

Numerous methods for alleviating catastrophic forgetting have been currently proposed. The most pragmatical way is to jointly train deep neural network models on both old and new tasks, which however demands a large amount of resources to store previous training data and hinders the learning of novel data in real time. Another option is to complement the training data for each new task with "pseudo-data" of the previous tasks Shin et al. (2017); Robins (1995). Besides the main model for task performance, a separate generative model is trained to generate fake historical data used for pseudo-rehearsal. Deep Generative Replay (DGR) Shin et al. (2017) replaces the storage of the previous training data with a Generative Adversarial Network to synthesize training data on all previously learnt tasks. These generative approaches have succeeded over very simple and artificial inputs but they cannot tackle more complicated inputs Atkinson et al. (2018). Moreover, to synthesize the historical data reasonably well, the size of the generative model is usually huge that costs much memory Wen et al. (2018). An alternative method is to store the weights of the model trained on previous tasks, and impose constraints of weight updates on new tasks He & Jaeger

(2018); Kirkpatrick et al. (2017); Zenke et al. (2017); Lee et al. (2017); Lopez-Paz et al. (2017). For example, Elastic Weight Consolidation (EWC) Kirkpatrick et al. (2017) and Learning Without Forgetting (LwF) Li & Hoiem (2018) store all the model parameters on previously learnt tasks, estimate their importance on previous tasks and penalize future changes to the weights on new tasks. However, selecting the "important" parameters for previous tasks complicates the implementation by exhaustive hyper-parameter tuning. In addition, state-of-the-art neural network models often involve millions of parameters and storing all network parameters from previous tasks does not necessarily reduce the memory cost Wen et al. (2018). In contrast with these methods, storing a small subset of examples from previous tasks and replaying the "exact subset" substantially boost performance Kemker & Kanan (2017); Rebuffi et al. (2017); Nguyen et al. (2017). To achieve the desired network behavior on previous tasks, incremental Classifier and Representation Learner (iCARL) Rebuffi et al. (2017) and Few-shot Self-Reminder (FSR) Wen et al. (2018) follow the idea of logit matching or knowledge distillation in model compression Ba & Caruana (2014); Bucilua et al. (2006); Hinton et al. (2015). However, such approaches ignore the topological relations among clusters in the embedding space and rely too much on a small amount of individual data, which may result in overfitting as shown in our experiments (Section 4.2). In contrast with them, without tampering the performance in memory retention, our method learns embedding functions and compares the feature similarities represented by class prototypes in the embedding space which improves generalization, especially in the few-shot settings, as also been verified in works Hoffer & Ailon (2015); Snell et al. (2017).

In this paper, we propose the method, Prototype Recalls, for continual learning in classification tasks. Similar as Snell et al. (2017), we use a neural network to learn class-representative prototypes in an embedding space and classify embedded test data by finding their nearest class prototype. To tackle the problem of catastrophic forgetting, we impose additional constraints on the network by classifying the embedded test data based on prototypes from previous tasks, which promotes the preservation of initial embedding function. For example (Figure 1), in the first task (Subfigure 1a), the network learns color prototypes to classify blue and yellow circles and in the second task (Subfigure 1b), the network learns shape prototypes to classify green circles and triangles. With catastrophically forgetting color features, the network extracts circle features on the first task and fails to classify blue and yellow circles. To alleviate catastrophic forgetting, our method replays the embeded previous samples (blue and yellow circles) and match them with previous color prototypes (blue and yellow) which reminds the network of extracting both color and shape features in both classification tasks.

We evaluate our method under two typical experimental protocols, incremental domain and incremental class, for continual learning across three benchmark datasets, MNIST Deng (2012), CIFAR10 Krizhevsky & Hinton (2009) and miniImageNet Deng et al. (2009). Compared with the state-of-the-arts, our method significantly boosts the performance of continual learning in terms of memory retention capability while being able to adapt to new tasks. Unlike parameter regularization methods or iCARL or FSR, our approach further reduces the memory storage by replacing logits of each data or network parameters with one prototype of each class in the episodic memory. Moreover, in contrast to these methods where the last layer in traditional classification networks often structurally depends on the number of classes, our method leverages on metric learning, maintains the same network architecture and does not require adding new neurons or layers for new object classes. Additionally, without sacrificing classification accuracy on initial tasks, our method can generalize to learn new concepts given a few training examples in new tasks due to the advantage of metric learning, commonly used in few-shot settings Snell et al. (2017); Hoffer & Ailon (2015).

## 2 PROPOSED METHOD

We propose the method, Prototype Recalls, for continual learning. For a sequence of datasets $D_1, D_2, ..., D_t, ...$, given $D_t$ in any task $t$ where $t \in \{1, 2, ..., T\}$, the goal for the model $f_T$ is to retain the good classification performance on all $T$ datasets after being sequentially trained over $T$ tasks. The value of $T$ is not pre-determined. The model $f_T$ with learnable parameters $\phi$ is only allowed to carry over a limited amount of information from the previous $T - 1$ tasks. This constraint eliminates the naive solution of combining all previous datasets to form one big training set for fine-tuning the model $f_T$ at task $T$. Each dataset $D_t$ consists of $N_t$ labeled examples $D_t = \{X_t, Y_t\} = \{(x_{1t}, y_{1t}), ..., (x_{Nt}, y_{Nt})\}$ where each $x_{it} \in \mathbb{R}^D$ is the $D$-dimensional feature

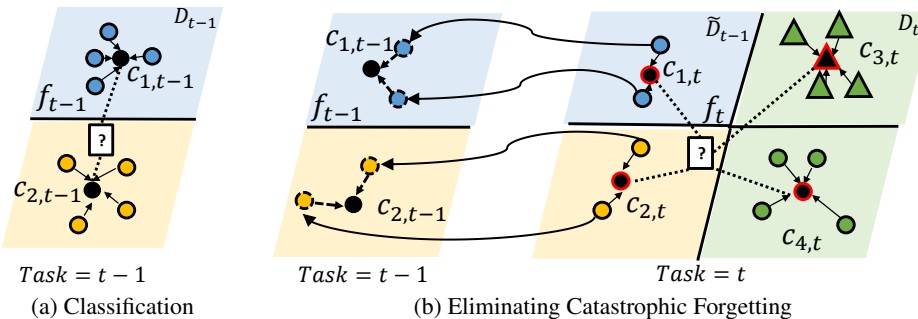

$$\text{Task} = t - 1 \qquad \text{Task} = t - 1 \qquad \text{Task} = t$$

(a) Classification          (b) Eliminating Catastrophic Forgetting

Figure 1: Illustration of classification and catastrophic forgetting alleviation in our proposed method. In **(a)**, dataset $D_{t-1}$ contains two classes. Prototypes $c_{1,t-1}$ and $c_{2,t-1}$ are obtained by averaging example embeddings of the same class. White square with question mark (embedded test data) can be classified by finding its nearest prototype. In **(b)**, dataset $D_t$ containing two new classes is introduced in task $t$. The prototypes $c_{i,t}, i = 1, \ldots, 4$ on task $t$ are computed from $D_t \bigcup \tilde{D}_{t-1}$ where $\tilde{D}_{t-1}$ is a randomly sampled subset from $D_{t-1}$. To eliminate catastrophic forgetting, our method constantly recalls metric space at task $t - 1$ by making example embeddings on $\tilde{D}_{t-1}$, as denoted by dashed color shapes, close to prototypes $c_{t-1}$.

vector of an example and $y_{it} \in \{1, ..., K_t\}$ is the corresponding class label. $S_{kt}$ denotes the set of examples labeled with class $k_t$.

At task $T$, if we simply train a model by only minimizing the classification loss $L_{classi, D_T}$ on dataset $D_T$, the model will forget how to perform classification on previous datasets $D_t, t < T$ which is described as catastrophic forgetting problem McCloskey & Cohen (1989); Ratcliff (1990); French (1999). Here we show how the model trained in our method retains the good performance on all previous tasks while adaptively learning new tasks. The loss for all the previous datasets is denoted by $L_T(f) = \sum_{t=1}^{T} \mathbb{E}_{D_t}[L(f(X_t), Y_t)]$. Our objective is to learn $f_T$ defined as follows:

$$f_T = \arg\min_f L_T(f) = \arg\min_f \{\sum_{t=1}^{T-1} L_{classi, D_t}(f_t) + L_{classi, D_T}(f) + \sum_{t=1}^{T-1} \delta_{D_t}(f, f_t)\} \quad (1)$$

where $L_{classi, D_T}(f)$ defines the classification loss of $f$ on dataset $D_T$ and $\delta_{D_t}(f, f_t)$ measures the differences in the network behaviors in the embedding space learnt by $f$ and $f_t$ on $D_t$, as introduced later in Equ 7. Given $f_1, ..., f_{T-1}$ that are learnt from the previous tasks, at task $T$, learning $f_T$ requires minimizing both terms $L_{classi, D_T}(f)$ and $\sum_{t=1}^{T-1} \delta_{D_t}(f, f_t)$. In the subsections below and Figure 1, we describe how to optimize these two terms.

## 2.1 CLASSIFICATION

To perform classification on dataset $D_t$, our method learns an embedding space in which points cluster around a single prototype representation for each class and classification is performed by finding the nearest class prototype Snell et al. (2017) (Figure 1a). Compared to traditional classification networks with a specific classification layer attached in the end, such as iCARL and FSR, our method keeps the network architecture unchanged while finding the nearest neighbour in the embedding space, which would lead to more efficient memory usage. For example, in one of the continual learning protocols Snell et al. (2017) where the models are asked to classify incremental classes (also see Section 3.1), traditional classification networks have to expand their architectures by accommodating more output units in the last classification layer based on the number of incremental classes and consequently, additional network parameters have to be added into the memory.

Without loss of generality, here we show how our method performs classification on $D_T$. First, the model learns an embedding function $f : \mathbb{R}^D \rightarrow \mathbb{R}^M$ and computes an $M$-dimensional prototype $c_{kT} \in \mathbb{R}^M$ which is the mean of the embeddings from examples $S_{kT}$:

$$c_{kT} = \frac{1}{|S_{kT}|} \sum_{(x_{iT}, y_{iT}) \in S_{kT}} f(x_{iT}). \quad (2)$$

The pairwise distance of one embedding and one prototype within the same class should be smaller than the intra-class ones. Our method introduces a distance function $d : \mathbb{R}^M \times \mathbb{R}^M \to [0, \infty)$. For each example $x_T$, it estimates a distance distribution based on a softmax over distances to the prototypes of $K_T$ classes in the embedding space:

$$p_\phi(y_T = k_T | x_T) = \frac{\exp(-d(f(x_T), c_{kT}))}{\sum_{k'}^{K_T} \exp(-d(f(x_T), c_{k'T}))}. \tag{3}$$

The objective function $L_{classi, D_T}(f)$ is to minimize the negative log-probability $-\log p_\phi(y_T = k_T | x_T)$ of the ground truth class label $k_T$ via Stochastic Gradient Descent Bottou (2010):

$$L_{classi, D_T}(f) = -\log p_\phi(y_T = k_T | x_T) \tag{4}$$

In practice, when $N_T$ is large, computing $c_{kT}$ is costly and memory inefficient during training. Thus, at each training iteration, we randomly sample two complement subsets from $S_{kT}$ over all $K_T$ classes: one for computing prototypes and the other for estimating distance distribution. Our primary choice of the distance function $d(\cdot)$ is squared Euclidean distance which has been verified to be effective in Snell et al. (2017). In addition, we include temperature hyperparameter $\tau$ in $d(\cdot)$ as introduced in network distillation literature Hinton et al. (2015) and set its value empirically based on the validation sets. A higher value for $\tau$ produces a softer probability distribution over classes.

## 2.2 PROTOTYPE RECALL

Regardless of the changes of the network parameters from $\phi_t$ to $\phi_T$ at task $t$ and $T$ respectively, the primary goal of $f_T$ is to learn the embedding function which results in the similar metric space as $f_t$ on dataset $D_t$ in task $t$ (Figure 1b). Given a limited amount of memory, a direct approach is to randomly sample a small subset $\tilde{D}_t = \{(x_i^{(t)}, y_i^{(t)}) | i = 1, ..., m\}$ from $D_t$ and replay these examples on task $T$. There have been some attempts Chen et al. (2012); Koh & Liang (2017); Brahma & Othon (2018) selecting representative examples for $\tilde{D}_t$ based on different scoring functions. However, the recent work Wen et al. (2018) has shown that random sampling uniformly across classes has already yielded outstanding performance in continual learning tasks. Hence, we adopt the same random sampling strategy to form $\tilde{D}_t$.

Intuitively, if the number of data samples in $\tilde{D}_t$ is very large, the network could re-produce the metric space at task $t$ by replaying $\tilde{D}_t$, which is our desired goal. However, this does not hold in practice given limited memory capacity. With the simple inductive bias that the metric space at task $t$ can be underlined by class-representative prototypes, we introduce another loss that embedded data sample in $\tilde{D}_t$ should still be closest to their corresponding class prototype among all prototypes at task $t$. This ensures the metric space represented by a set of prototypes learnt from $\tilde{D}_t$ by $f_T$ provides good approximation to the one in task $t$.

Formally, for any $f$ after task $t$, we formulate the regularization of network behaviors $\delta_{D_t}(f, f_t)$ in the metric space of task $t$ by satisfying two criteria: first, $f$ learns a metric space to classify $\tilde{D}_t$ by minimizing the classification loss $L_{classi, \tilde{D}_t}(f)$, as introduced in Sec. 2.1 above; second, to preserve the similar topological structure $L_{regu, \tilde{D}_t, D_t}(f, f_t)$ among clusters on dataset $D_t$, the embeddings $f(\tilde{x}_t)$ predicted by $f$ based on $\tilde{D}_t$ should produce the similar distance distribution based on a softmax over the distance to prototypes $c_{kt}$ computed using $f_t$ on dataset $D_t$:

$$p_\phi(\tilde{y}_t = \tilde{k}_t | \tilde{x}_t) = \frac{\exp(-d(f(\tilde{x}_t), c_{kt}))}{\sum_{k'}^{K_t} \exp(-d(f(\tilde{x}_t), c_{k't}))}, \quad c_{kt} = \frac{1}{|S_{kt}|} \sum_{(x_{it}, y_{it}) \in S_{kt}} f_t(x_{it}). \tag{5}$$

Concretely, $L_{regu, \tilde{D}_t, D_t}(f, f_t)$ is to minimize the negative log-probability $p_\phi(\tilde{y}_t = \tilde{k}_t | \tilde{x}_t)$ of the ground truth class label $\tilde{k}_t$ conditioned on prototypes $c_{kt}$, which is pre-computed using $f_t$ in Eq 5 at task $t$ and stored in the episodic memory until task $T$:

$$L_{regu, \tilde{D}_t, D_t}(f, f_t) = -\log p_\phi(\tilde{y}_t = \tilde{k}_t | \tilde{x}_t). \tag{6}$$

Overall, we define $\delta_{D_t}(f, f_t)$ in Eq 1 as below:

$$\delta_{D_t}(f, f_t) = L_{classi, \tilde{D}_t}(f) + L_{regu, \tilde{D}_t, D_t}(f, f_t). \tag{7}$$

---

**Algorithm 1:** Prototype recall algorithm in continual learning for a training episode

---

**Input** : A sequence of datasets $D_1, D_2, ..., D_t, ...$, one per task $t$. A feed-forward neural network learning embedding function $f$. Episodic memory with capacity $C$. Sampled $m$ examples per dataset.

**Output:** Update the network parameters $\phi_t$

**for** *each task t* **do**

    **if** $t = 1$ **then**

        Classify $D_1$; Compute prototypes $\{c_{11}, ..., c_{K1}\}$ ;             `// Equ. 2, 3, 4`

        Store prototypes $\{c_{11}, ..., c_{K1}\}$ and $m$ sampled examples $\tilde{D}_1$ in episodic memory;

    **else**

        Classify $D_t$; Compute prototypes $\{c_{1t}, ..., c_{Kt}\}$ ;             `// Equ. 2, 3, 4`

        **for** *all previous tasks i from* $1$ *to* $t - 1$ **do**

            minimize $\delta_{D_t}(f, f_i)$ using the following ;              `// Equ. 7`

            Classify $\tilde{D}_i$ ;                              `// Equ. 2, 3, 4`

            Prototype recalls using $\tilde{D}_i$ and prototypes $\{c_{1i}, ..., c_{Ki}\}$ in episodic memory ; `// Equ. 5, 6`

        Update $m$ based on $C$; Store prototypes $\{c_{1t}, ..., c_{Kt}\}$ and $\tilde{D}_t$ in episodic memory ;    `// Equ. 8`

---

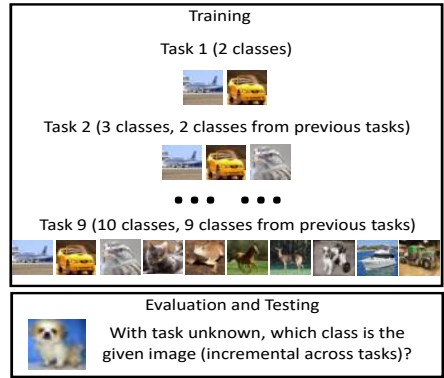

| Training | Training |
|---|---|
| Task 1 (permutation 1) | Task 1 (2 classes) |
| Task 2 (permutation 2) | Task 2 (3 classes, 2 classes from previous tasks) |
| • • •   • • • | • • •   • • • |
| Task 20 (permutation 20) | Task 9 (10 classes, 9 classes from previous tasks) |
| Evaluation and Testing | Evaluation and Testing |
| With permutation unknown, which digit is the given image (from 0 to 9)? | With task unknown, which class is the given image (incremental across tasks)? |

(a) Permuted MNIST in Incremental Domain        (b) Split CIFAR10 in Incremental Class

Figure 2: Schematics of two task protocols in our continual learning experiments: (a) learning with incremental domain on permuted MNIST; and (b) learning with incremental classes on split CIFAR10.

### 2.3 DYNAMIC EPISODIC MEMORY ALLOCATION

Given a limited amount of memory with capacity $C$, our proposed method has to store a small subset $\tilde{D}_t$ with $m$ examples randomly sampled from $D_t$ and prototypes $c_{kt}, k \in \{1, ..., K_t\}$ computed using embedding function $f_t : \mathbb{R}^D \to \mathbb{R}^M$ on $D_t$ where $t < T$. The following constraint has to be satisfied:

$$C = \sum_{t=1}^{T-1} K_t(M + mD) \tag{8}$$

When the number of tasks $T$ is small, $m$ can be large and the episodic memory stores more examples in $\tilde{D}_t$. Dynamic memory allocation of enabling more example replays in earlier tasks puts more emphasis on reviewing earlier tasks which are easier to forget, and introduces more varieties in data distributions when matching with prototypes. Pseudocode to our proposed algorithm in continual learning for a training episode is provided in Algorithm 1.

## 3 EXPERIMENTAL DETAILS

We introduce two task protocols for evaluating continual learning algorithms with different memory usage over three benchmark datasets. Source codes will be public available upon acceptance.

## 3.1 TASK PROTOCOLS

**Permuted MNIST in incremental domain task** is a benchmark task protocol in continual learning Lee et al. (2017); Lopez-Paz et al. (2017); Zenke et al. (2017) (Figure 2a). In each task, a fixed permutation sequence is randomly generated and is applied to input images in MNIST Deng (2012). Though the input distribution always changes across tasks, models are trained to classify 10 digits in each task and the model structure is always the same. There are 20 tasks in total. During testing, the task identity is not available to models. The models have to classify input images into 1 out of 10 digits.

**Split CIFAR10 and split MiniImageNet in incremental class task** is a more challenging task protocol where models need to infer the task identity and meanwhile solve each task. The input data is also more complex which includes classification on natural images in CIFAR10 Krizhevsky & Hinton (2009) and miniImageNet Deng et al. (2009). The former contains 10 classes and the latter consists of 100 classes. In CIFAR10, the model is first trained with 2 classes and later with 1 more class in each subsequent task. There are 9 tasks in total and 5,000 images per class in the training set. In miniImageNet, models are trained with 10 classes in each task. There are 10 tasks in total and 480 images per class in the training set.

**Few-shot Continual Learning** Humans can learn novel concepts *given a few examples* without sacrificing classification accuracy on initial tasks Gidaris & Komodakis (2018). However, typical continual learning schemes assume that a large amount of training data over all tasks is always available for fine-tuning networks to adapt to new data distributions, which does not always hold in practice. We revise task protocols to more challenging ones: networks are trained with a few examples per class in sequential tasks except for the first task. For example, on CIFAR10/miniImageNet, we train the models with 5,000/480 example images per class in the first task and 50/100 images per class in subsequent tasks.

## 3.2 BASELINES

We include the following categories of continual learning methods for comparing with our method. To eliminate the effect of network structures in performance, we introduce control conditions with the same architecture complexity for all the methods in the same task across all the experiments.

**Parameter Regularization Methods**: Elastic Weight Consolidation (EWC) Kirkpatrick et al. (2017), Synaptic Intelligence (SI) Zenke et al. (2017) and Memory Aware Synapses (MAS) Aljundi et al. (2018) where regularization terms are added in the loss function; online EWC Schwarz et al. (2018) which is an extension of EWC in scalability to a large number of tasks; L2 distance indicating parameter changes between tasks is added in the loss Kirkpatrick et al. (2017); SGD, which is a naive baseline without any regularization terms, is optimized with Stochastic Gradient Descent Bottou (2010) sequentially over all tasks.

**Memory Distillation and Replay Methods**: incremental Classifier and Representation Learner (iCARL) Rebuffi et al. (2017) and Few-shot Self-Reminder (FSR) Wen et al. (2018) propose to regularize network behaviors by exact pseudo replay. Specifically, in FSR, there are two variants: FSR-KLD for logits matching via KullbackLeibler Divergence loss and FSR-MSE for logits distillation via L2 distance loss.

Performance is reported in terms of both mean and standard deviation after 10 runs per protocol. Since generative model-based approaches van de Ven & Tolias (2018); Shin et al. (2017) greatly alter architecture of the classification networks, we do not compare with them.

## 3.3 MEMORY COMPARISON

For fair comparison, we use the *same* feed-forward architecture for all the methods and allocate a comparable amount of memory as EWC Kirkpatrick et al. (2017) and other parameter regularization methods, for storing example images per class and their prototypes. In EWC, the model often allocates a memory size twice as the number of network parameters for computing Fisher information matrix which can be used for regularizing changes of network parameters Kirkpatrick et al. (2017). In more challenging classification tasks, the network size tends to be larger and hence, these methods require much more memory. In Table 1, we show an example of memory allocation

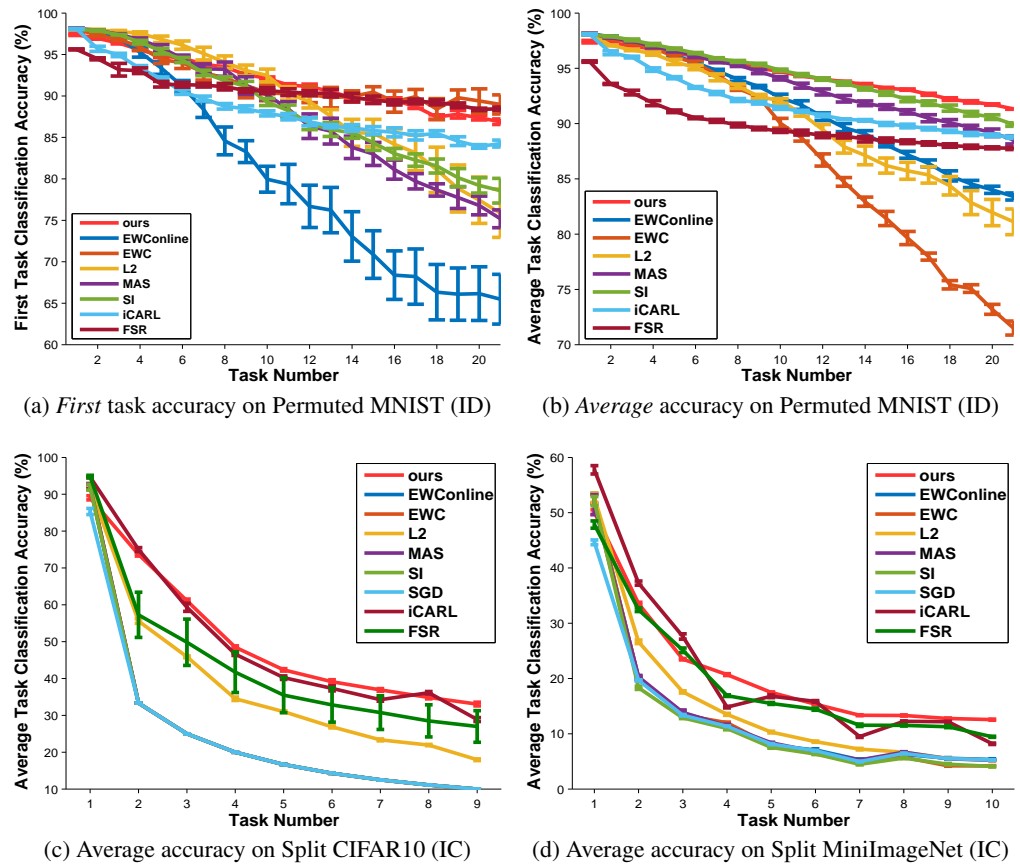

Figure 3: First and average task classification accuracies from 10 random runs on permuted MNIST, split CIFAR10 and split miniImagetNet in incremental domain (ID) and incremental class (IC) tasks.

on split CIFAR10 in incremental class tasks with full memory and little memory respectively. The feed-forward classification network contains around $16.3 \times 10^5$ parameters. Weight regularization methods require memory allocation twice as that, which takes about $32.63 \times 10^5$ parameters. The input RGB images are of size $3 \times 32 \times 32$. Via Equ. 8, our method can allocate episodic memory with full capacity $C = 16.3 \times 10^5$ and calculate $m$ which is equivalent to storing $16.3 \times 10^5 / (3 \times 32 \times 32) = 530$ example images per class. In experiments with little training data as described in Section 3.1, we reduce $m$ to 10 example images per class.

## 4 EXPERIMENTAL RESULTS

### 4.1 ALLEVIATING FORGETTING

Figure 3 reports the results of continual learning methods with full memory under the two task protocols. All compared continual learning methods outperform SGD (cyan) which is a baseline without preventing catastrophic forgetting. Our method (red) achieves the highest average classification accuracy among all the compared methods, including both parameter regularization methods and memory-based methods, with minimum forgetting.

A good continual learning method should not only show good memory retention but also be able to adapt to new tasks. In Figure 3a, although our method (red) performs on par with EWC (brown) and FSR (date) in retaining the classification accuracy on dataset $D_1$ in the first task along with 20 sequential tasks, the average classification accuracy of our method is far higher than EWC (brown) and FSR (date) as shown in Figure 3b, indicating both of these methods are able to retain good memory but fail to learn new tasks. After the 13th task, the average classification performance of

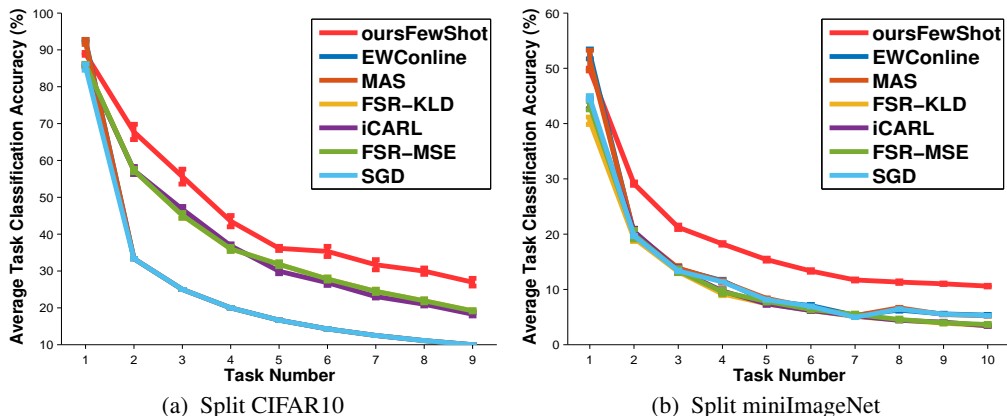

Figure 4: Few-shot average task classification accuracies on split CIFAR10 and miniImagetNet in incremental class task with few training data in the second tasks and onwards.

EWC is even worse than SGD. Across total 20 tasks, our method leads FSR (date) by 3% more accurate on average. Similar reasoning can be applied to comparison with SI (green): although our method performs comparably well as SI in terms of average classification accuracy, SI fails to retain the classification accuracy on $D_1$, which is 6% lower than ours in the 20th task.

Figure 3c and 3d show the average task classification accuracy over sequential tasks in incremental class protocol. Incremental class protocol is more challenging than incremental domain protocol, since the models have to infer both the task identity and class labels in the task. Our method (red) performs slightly better than iCARL (date) and has the hightest average classification accuracy in continual learning. Compared with third best method, FSR (green), our method yields constantly around 5% higher on average across all tasks on CIFAR10 and miniImageNet respectively. Note that most weight regularization methods, such as EWC (brown), perform as badly as SGD. It is possible that EWC computes Fisher matrix to maintain local information and does not consider the scenarios when data distributions across tasks are too far apart. On the contrary, our method maintains remarkably better performance than EWC, because ours focuses primarily on the behaviors of network outputs, which indirectly relaxes the constraint about the change of network parameters.

## 4.2 FEW-SHOT CONTINUAL LEARNING

We evaluate continual learning methods with little memory under two task protocols with few training data in the second tasks and onwards except for the first tasks. Figure 4 reports their performance. Our method (red) has the highest average classification accuracy over all sequential tasks among state-of-the-art methods with 27% and 11% vs. 19% and 4% of FSR-KLD (yellow), which is the second best, at the 9th and 10th tasks on CIFAR10 and miniImageNet respectively. Weight regularization methods, such as EWConline (blue) and MAS (brown), perform as badly as SGD (cyan), worse than logits matching methods, such as FSR (green and yellow) or iCARL (purple). Similar observations have been made as Figure 3 with full training data.

Compared with logits matching methods, our method has the highest average task classification accuracy. It reveals that our method performs classification via metric learning in an effective few-shot manner. It is also because our network architecture is not dependent on the number of output classes and the knowledge in previous tasks can be well preserved and transferred to new tasks. It is superior to traditional networks with new parameters added in the last classification layer, which easily leads to overfitting. As a side benefit, given the same number of example inputs in the episodic memory, our method is more efficient in memory usage since it stores one prototype per class instead of the logits for each example input as verified in Table 1.

## 4.3 NETWORK ANALYSIS

We also study the effects of the following three factors upon performance improvement. Figure 5 reports the average classification accuracy of these ablated methods. (1) Intuitively, limited memory capacity restricts number of example inputs to re-play and leads to performance drop. On permuted MNIST in incremental domain, with full memory capacity reduced by 2.5 times (from 5,000

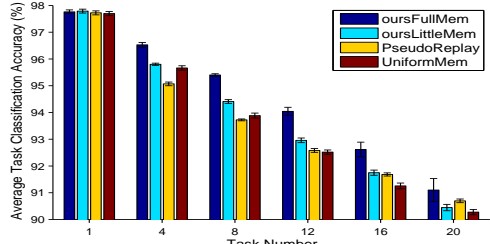

Figure 5: Average task classification accuracies for the ablated models on permuted MNIST in incremental domain task.

| Full Training and Full Memory Size in Magnitudes of $10^5$ | | | | | |
|---|---|---|---|---|---|
| EWC | EWC online | MAS | SI | L2 | ours |
| 32.63 | 32.63 | 32.63 | 32.63 | 32.63 | **32.60** |
| Little Training and Little Memory Size in Magnitudes of $10^5$ | | | | | |
| MAS | EWC online | FSR-KLD | iCARL | FSR-MSE | ours |
| 32.63 | 32.63 | 19.40 | 19.40 | 19.40 | **19.39** |

Table 1: Memory allocation for continual learning methods on CIFAR10 in incremental class task.

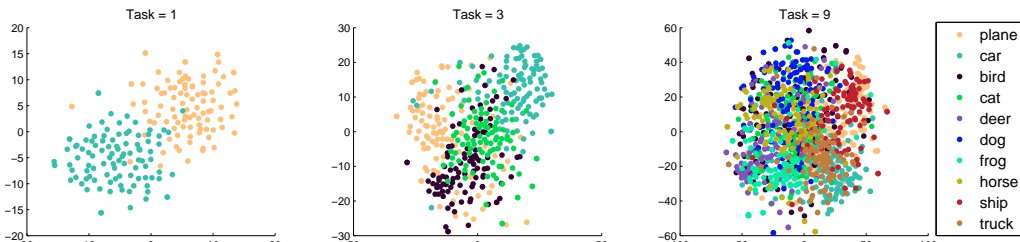

Figure 6: 2D visualization of embedding clusters learnt by our method using t-sne Van Der Maaten (2014) on split CIFAR10 in incremental class task. The first task (Task = 1) is a binary classification problem. In subsequent tasks, one new class is added.

example inputs to 2,000), our method shows a moderate decrease of average classification accuracy by 1% in the 20th task. (2) We also compare our method with memory replay optimized by cross-entropy loss at full memory conditions. A performance drop around 1.5% is observed which validates classifying example inputs based on initial prototypes results in better performance in memory retention. (3) Given fixed $C$, our method adopts the strategy of decreasing $m$ numbers of example inputs in memory, with the increasing number of tasks. The performance drop of 1.5% using uniform memory allocation demonstrates the usefulness of dynamic memory allocation which enforces more examples to be replayed in earlier tasks, and therefore promotes memory retention.

In Figure 6, we provide visualizations of class embeddings by projecting these latent representations of classes into 2D space. It can be seen that our method is capable of clustering latent representations belonging to the same class and meanwhile accommodating new class embeddings across sequential tasks. Interestingly, the clusters are topologically organized based on feature similarities among classes and the topological structure from the same classes is preserved across tasks. For example, the cluster of "bird" (black) is close to that of "plane" (orange) in Task 3 and the same two clusters are still close in Task 9. This again validates that classifying example inputs from previous tasks based on initial prototypes promotes preservation of topological structure in the initial metric space.

## 5 CONCLUSION

We address the problem of catastrophic forgetting by proposing prototype recalls in classification tasks. In addition to significantly alleviating catastrophic forgetting on benchmark datasets, our method is superior to others in terms of making the memory usage efficient, and being generalizable to learning novel concepts given only a few training examples in new tasks.

However, given a finite memory capacity and a high number of tasks, we recognize that our method, just like other memory-based continual learning algorithms, have limitations in number of prototypes stored. The memory requirement of our method increases linearly with the number of continuous tasks. In practice, there is always a trade-off between memory usage and retention. We believe that our method is one of the most efficient continual learning methods in eliminating catastrophic forgetting with a decent amount of memory usage. Moreover, we restrict ourselves in classification tasks with discrete prototypes. In the future work, to apply our algorithm in more complex and challenging problems, such as regression and reinforcement learning (RL), one possible solution is to quantize the continuous space in regression or formulate RL in discrete state-action pairs.

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
