# OpenReview forum: "Prototype Recalls for Continual Learning"
_ICLR.cc/2020/Conference — Reject_

### Official Review · AnonReviewer1 · 2019-10-22
**Official Blind Review #1**

**Rating:** 3

**Review:**

Paper proposes a method for continual learning. The method is based on the learning of a metric space where classes are represented by prototypes in this space. To prevent forgetting the method proposes to perform prototype recall, aiming to keep prototypes in the same location in embedding space (Fig 1b). The method is compared with several recent methods and is shown to outperform them on two small datasets (MNIST permuted and CIFAR10). The idea of using prototypes for continual learning is interesting, as the authors point out, this does not require adding new neurons to the network for new tasks.

I found the paper hard to follow (presentation needs improvement) and the experimental results are lacking. I, therefore, recommend weak reject at the moment.

Clarity method:
I did an effort to understand the details of the paper but found the chosen notation hard to follow. Some remarks with respect to that.

1. The delta loss is based on previous models (last term Eq. 1), which you do not want to store. It is important to immediately point out that you are not storing them (if I understand correctly) and that you are actually only storing the prototypes (Eq. 5).
2. Could the loss in Eq 6 be seen as a distillation loss for an embedding space. You want to stay close to previous network ?
3. It was unclear for me how often prototypes where updated, in every minibatch ?
4.The algorithm is not very clear (what is a training episode?). Sampled m examples per dataset (should be per class per dataset). From the algorithm it seems that the loss in Eq 4 (first line after else) is first minimized separately from the loss in Eq. 7 . Should this not been done jointly ?
5.What is the contribution of dynamic episodic memory allocation ? I am not sure if the Equation is very helpful. Is in practice M<<D ?
6. Exemplars can only be chosen ones (when considering the task). I would like the authors to confirm  that they do that, because I was not sure from the text (they can be deleted later but not replaced with other exemplars from the same task).

Experimental results:
1. The authors should provide more experimental details. What network is used? Is it a pretrained network.
2. Colors in Fig 3 preferably stay the same for same methods. The legend should only show results which are present in graph. (Fig 3c has 5 curves and 10 methods in legend). Why are there 21 tasks in Fig 3a and b, should it not be 20 ? Why are there 9 tasks in 3c,  should it not be 10 ? SGD results missing in Fig 3a and b, but mentioned in text. What happens with EWC, it has excellent results in Fig 3a but does bad in 3b, is there an explanation ?
3. Ablation study should include ablation of the proposed losses, do they actually help in improving the results. The ablation is on memory usage which is not the main contribution of the paper. It would  be better to use CIFAR 10 then MNIST premuted for ablation (see for example ‘A COMPREHENSIVE, APPLICATION-ORIENTED STUDY OF CATASTROPHIC FORGETTING IN DNNS’ on the limitations of MNIST permuted).
4. Figure 6 did not help me understnad the method. It would be nice if you could plot the prototypes, than we can see if they keep in the same location.

Minor:
I am not sure the authors are aware of difference \cite and \citep which would improve readability.


**Experience Assessment:**

I have published one or two papers in this area.

**Review Assessment: Checking Correctness Of Derivations And Theory:**

I assessed the sensibility of the derivations and theory.

**Review Assessment: Checking Correctness Of Experiments:**

I carefully checked the experiments.

**Review Assessment: Thoroughness In Paper Reading:**

I read the paper thoroughly.

---

### Official Review · AnonReviewer2 · 2019-10-22
**Official Blind Review #2**

**Rating:** 1

**Review:**

The proposed method addresses continual learning, by learning a mapping from the input space to an embedding space, and employing a loss that encourages clustering the embeddings by class (and task?) around some centroids called prototypes. Catastrophic forgetting is mitigated by adding a penalty term that is proportional to the distance of the embeddings under the current network of some samples from the past tasks, and the centroids previously associated to each of them.

The idea might have some merit, but the paper in its current state fails to convey it clearly and is too vague on the experimental settings for the results to be considered in any way. The notation is at times unnecessary heavy (see specific comments below), and many details on the method and experimental setup are missing to the point that it’s difficult to judge it and it would be impossible to reproduce it. In addition, it’s unclear whether the baselines have been properly tuned since the reported performance is incompatible with that of the original paper.

The submitted paper is clearly not ready for publication. I can consider modifying my vote if the paper undergoes a major review, fixing the notation, explaining the method and the experimental settings extensively and clearly, with proper comparison in terms of capacity/performance tradeoff w.r.t. the state of the art.


Detailed feedback:

1) Notation
- I believe the exposition would be more fluent and easier to follow if you described the method under the assumption that all tasks have the same number of classes and a shared dataset, and mentioning that the method can be easily extended to work in the more general setting. Under this assumption, e.g., c_{kT} would simplify to c_{k}.

- When you refer to a generic task, please use a lower case letter (e.g., t) instead of T (e.g., first paragraph of page 3, Eqn. 1, Eqn. 2, ..). Use capital letters for constants only (e.g., use T for the total number of tasks). When you need to refer to two tasks that are in a particular relative ordering. For instance, Eqn. 1 should be the argmin of “\sum_{s=1}^{t} [...], with s < t”. Similarly, Sec 2.2 should refer to tasks s and t, rather than t and T (unless T is specifically referring to the last task alone, which doesn’t seem the case here as from what I gather the same should hold for all tasks, i.e., for a specific task t and any previous task s, with s < t.).

- If I am not mistaken, L_T is defined just before Eqn. 1 and only used there. It is not very informative and it’s based on some undefined loss L. I would suggest to drop it entirely, and simplify Eqn. 1 accordingly.

- Please avoid subscript with commas as much as possible, as they are difficult to parse. E.g., L_{classi, Dt} can become L_{classi_t} (or even L_{c_t} or L_{c}^{(t)}), remaining equally informative. All other “D_t” subscripts can be similarly replaced by “t” throughout the text. Generally speaking, I would recommend the ^{(t)} notation to denote task t (which you already adopted when defining \tilde{D}_t

- I believe Eqn. 1 can be simplified by merging the two summations into a single expression. Also, I think it would be clearer to define the argmin to be over “f \in \mathcal{F}”, since it is a search over the space of all possible functions. Finally, I also find a bit confusing the use of f and f_t to denote an evolving (currently evaluated) function and a fixed (best function from previous search) one respectively, but I am not sure I have a suggestion to improve this notation.

- Please avoid unnecessary repetitions. In a number of cases a big part of a formula is defined inline in the text just before the equation. This doesn’t add to the conversation and makes it unnecessarily hard to parse the text (e.g., inline formulas just before Eqns. 4 and 6. Also prefer a textual description instead of the variable when introducing a definition (e.g., before Eqn. 7, “we define the distance as” or “we define the distance between the functions f and f_t embeddings as”)

- Please fix all citations according to the ICLR style guidelines: When the authors or the publication are included in the sentence, the citation should not be in parenthesis (as in ``See \citet{Hinton06} for more information.''). Otherwise, the citation should be in parenthesis (as in ``Deep learning shows promise to make progress towards AI~\citep{Bengio+chapter2007}.''). For instance, all citations in Sec 2.2 should use \citep.

2) To my best understanding classification in the embedding space is performed with some variation of k-means, with k being the number of “prototypes” for the task at hand (i.e., the number of classes for the specific task). If this is correct, referring to k-means in the paper and highlighting the similarities and differences to it, would have been a more effective way to describe your method.

3) The authors claim not to use a final classification, as opposed to competing methods, hence not being subject to memory usage increases when the number of classes increase (sec 2.1). Surprisingly though, they then employ a softmax over the prototypes (Eqn. 3) which to the best of my knowledge would require an incoming layer whose dimensionality would need to grow with the increase of the number of prototypes, (i.e., classes). Please update the text to either correct the claim or explain exactly how your method can drive the softmax in the case of an increasing number of classes without increasing the dimensionality of the layer before the softmax.

4) Similarly, the authors claim that “Unlike parameter regularization methods or iCARL or FSR, our approach further reduces the memory storage by replacing logits of each data or network parameters with one prototype of each class in the episodic memory”. This is not entirely true, as they also need to store a (unspecified) number of samples from all tasks to prevent forgetting. It is unclear that replacing a snapshot of the network parameters is better (in terms of memory consumption) than storing some data from each task, and a proper discussion on this crucial point of the paper is missing.


5) Apart from the losses, all the details on the architecture (type and number of layers, activation functions, ..) and on the hyperparameters are missing. The paper only mentions that the same feedforward architecture as EWC is used, with similar memory usage, but also mentions that unlike all other methods (including EWC I assume), theirs doesn’t need to vary the architecture if the number of tasks is increased. It is then completely unclear to me how this is possible if the architecture is the same. If it’s not, which parts of the network are shared and which are task specific. I assume most of the network to be shared among tasks and only the last layer (that drives the softmax that determines the probabilities of each class (prototype) for the current task) to be task-specific. This details are crucial to the understanding of the method and to be able to reproduce the results.

6) Crucially, the reported Permuted MNIST performance on EWC is much worse than that on the original paper. This suggests that the baselines have not been properly tuned or there are issues in the implementation.

7) Unless I am mistaken, in Sec 3.3 there seems to be a flaw in the calculation of the number of examples that can be stored by the proposed method in order to use approximately the same memory as the baselines. Indeed, the memory allocation for weight regularization is independent of the number of classes and so should be the equivalent capacity allocated for samples retention, i.e., in order to be comparable with EWC (and similar methods), the proposed model should be allowed to store 530 examples in total, rather than per class.

8) I suggest to drop the definitions of the obvious functions, such as the softmax and the NLL (that is even defined twice!), and use that space to improve the description of the model and algorithm instead.

9) Can you clarify what you mean by “The pairwise distance of one embedding and one prototype within the same class should be smaller than the intra-class ones.”? Should it be inter-class?

10) In my opinion the narrative would be clearer backwards: rather than first defining the loss as a sum of three sub-losses, I’d suggest to define the sub-losses first and finally aggregate them. In its current state, Eqn. 1 is not informative until \delta_{D_t} is defined, a full page later.

11) Eqns. 3 and 5 are not “distance distributions” but rather “class probabilities distributions”, if I understood correctly. Can you confirm and amend the text?

12) Eqn. 5 is more confusing than informative, I recommend to remove it and replace it with a better textual description. There is no need to redefine c_{kt}, as it is already defined in Eqn. 2 exactly in the same way. Secondly, there is no need to define the softmax over the distances a second time, as it’s already defined in Eqn. 3 and only the arguments change.

13) Re Sec 2.3, to my best understanding there is no “emphasis on reviewing earlier tasks”, as the number of rehearsal samples is the same for all tasks. In other words, \tilde{D}_t is sampled uniformly across tasks I believe. Is this right?

14) Sec 3.1, “typical continual learning schemes assume that a large amount of training data over all tasks is always available for fine-tuning”. This is not true, and it’s actually usually not the case  (see e.g., generative replay approaches, or those that rely on pruning or other forms of architecture modification.)

15) The introduction fails to mention the category of approaches that try to combat forgetting by being smart about the architecture, e.g., Progressive nets, Progress & compress, etc, .. Please amend this.



Minor:

- Add more space after Figure 1’s caption, so that it’s clear where the caption ends and the text begins.
- Eqn. 2, no need to define yiT since it’s not used in the equation.
- The sentence at the end of Sec 2.1 (“In practice … estimating the distance distribution” ..) should go after Eqn. 2. Similarly, the sentence “Our primary choice of  the distance function” would be better placed right after the definition of the distance function IMO.
- The MNIST citation is wrong. Please correct it: LeCun, Yann and Cortes, Corinna. "MNIST handwritten digit database." (1998)
- The introduction is a bit long and a “related work” section is missing. You might want to split the introduction into two chapters.
- Do not use D to denote the datasets and the dimensionality of the datasets.
- The sentence after Eqn. 1 “learning f_T requires minimizing both terms ..” is incorrect, as it goes against Eqn. 1 in which there are 3 terms rather than 2.
- End of page 3: “without loss of generality”...w.r.t. what? The phrase doesn’t seem to fit the context, consider changing it.
- End of page 7, the detailed description of Figure 3 would be better placed in the caption of the figure itself.

Typos:
- Page 2: state-of-the-arts -> state-of-the-art
- “There have been some attempts [citations] selecting”, “of” is missing
- End of page 4: which is pre-computed -> are
- End of page 5: public -> publicly

**Experience Assessment:**

I have read many papers in this area.

**Review Assessment: Checking Correctness Of Derivations And Theory:**

N/A

**Review Assessment: Checking Correctness Of Experiments:**

I carefully checked the experiments.

**Review Assessment: Thoroughness In Paper Reading:**

N/A

---

### Official Review · AnonReviewer3 · 2019-10-25
**Official Blind Review #3**

**Rating:** 3

**Review:**

Paper summary:
This paper proposes a method for continual learning/few-shot learning which involves (1) putting class prototypes in a replay buffer, at test time computing the distance to these prototypes in order to classify the example (2) regularizing the embedding space used to compute prototypes by ensuring classification accuracy on a replay buffer of data examples does not degrade.

Paper contributions:
 - The idea of putting prototypes in a replay buffer, and of the regularization for ensuring classification accuracy on earlier examples does not degrade (I'm not up-to-date enough in this area to know if these ideas are novel)
 - Experiments in continual learning and few shot learning which compare the proposed method to alternatives
 - Ablation study with different settings of the proposed method - exploration of the embedding clusters learned by the method

Review summary & decision:
This paper introduces two ideas that I think seem quite good and could be valuable to the community. However, I found the presentation of material very confusing and poorly organized, some claims misleading, and some experiments missing. I recommend rejection of the paper in its current state.

Reasons for decision:
1. The intro, "proposed method" section, and "prototype recall" section are repetitive of each other (especially proposed method and prototype recall). The latter two sections also have some information that should go in the related work (aka introduction). I actually like the merging of related work and introduction though; I think it could be nicely space-efficient if things weren't so repetitive later.
2. Overall I found the paper is very disorganized and hard to follow; the description of what is done (in particular the fact that you evaluate on three different tasks) is very unclear. It's mentioned in the last paragraph of the intro that you do two experimental protocols, but these are not referred to (at least not by the same name) in all of the experiments. Also there are experiments on few shot learning which are only ambiguously referred to in the abstract. The relationship of "classification" and "eliminating catastrophic forgetting" to these tasks is not discussed, even though there is a whole section called "classification" which seems like a misleading title for this section - it seems more like this is further description of your proposed method, along with some related work. The terms "single head" and "multi head" evaluation are in common use in continual learning, but are not referred to here at all. The pseudocode algorithm with references to equations makes it reasonably clear what is going on, but it would be helpful if it appeared much earlier and the discussion followed this outline (e.g. have this be the first thing in the section, and then have sections called "Prototype computation", "Episodic memory" and "Prototype recall"), and if it were made more clear where the actual task comes in. Also there's a lot of related work in the Experiments section. It seems like the few-shot experiments are kind of tacked on as an afterthought; the intro doesn't mention these and there are methods (e.g. logit matching) used here that are not mentioned in the intro/related work.
3. What is the motivation for averaging embeddings? As far as I can tell this is mentioned in the caption of Figure 1; it should be discussed elsewhere.
4. Part of the motivation for your approach as given in the abstract and intro is not to have to save examples, but your method does require this for computing the regularization term. I would like to see an ablation study of the proposed method with varying amounts of data examples, including 0 (i.e. without this regularization); as far as I understand this is only done for 10 and 530 examples. It seems odd to refer to this as "memory"; this only makes sense if we know you're referring to episodic memory, which is not clear (or even mentioned) in the caption for Figure 5. Based on the fact that the method performs poorly with only 10 examples as the number of tasks goes up, I would say it's misleading to claim you use only a small number of samples, as well as to claim that you only need to store class-representative prototypes (you also need to store these examples).

Feedback/suggestions/nits (not necessarily part of decision assessment):
 - "continual learning is a critical ability" this kind of subjective assessment doesn't add to our understanding and is generally not recommended in scientific writing. This first sentence does not do a good job of explaining what is continual learning for someone who does not know.
 - "enabling CL" strange word to use here.
  - Inconsistent capitalization of prototype recalls
 - In the abstract, the 4 contributions seem repetitive (1 and 3 seem the same to me) and the 4th contribution makes it sound like an avenue for future work than an actual contribution.
 - Intro is repetitive of the abstract
 - citations are frequently incorrectly formatted (should use \citep, not \citet unless you are referring to the authors by name)
 - unclear what "stationary batches" means; do you mean in continual learning the data distribution is non-stationary?
 - verb tense: "have been currently proposed" does not make sense
 - I don't understand the criticism of "Learning without forgetting" (that it requires exhaustive hyperparameter tuning); your method also has hyperparameters to tune. Do they have more, and is their method more sensitive to hyperparameters? What's the evidence for this?
 - If your criticism of replay-based methods is that they ignore topology among clusters in embedding space, you should explain why/how your method does not do this. Likewise for relying on "small amounts of individual data"; if seems to me that your method has the same structure, and for both your and these methods the size of the replay buffer is just a hyperparameter, unless I've misunderstood something.
 - "our method learns embedding functions... has also been verfied in works" what exactly has been verified? What do those works do? This sentence is unclear.
 - Hoffer & Ailon is incorrectly cited as Ailon (2015); in general the references are poorly formatted
 - Snell et al is barely mentioned in the related work, but the two methods seem very similar to me; I think it merits more discussion (in particular, highlighting how what you do is different / an extension). The "classification" section seems to imply you compute prototypes exactly as in Snell et al., but elsewhere it sounds like you do something different.
 -  Figure 1 is not clear; one must have already read the full explanation of the two settings (which is spread over the intro and other sections) to understand it. Just mentioning that these two are different evaluation settings would be helpful. Also,  I find the relationship between the tasks unclear.
  - "yielded outstanding performance" what is outstanding in this context?
 - not sure it's accurate to call "the metric space at task t can be underlined by class-representative prototypes" an "inductive bias". Also what does "underline" mean here?
 - "without loss of generality" I don't think this phrase is used appropriately here
 - Algorithm 1's for loop is quite unclear to me, in particular it's unclear where you actually do the task you're supposed to be doing.
 - EWC and yours are the same colour in Figure 3
 - Figure captions should mention what the abbreviations mean (or mention where in the text these explanations can be found).
 - What are we supposed to tak away from the TSNE plots? This explanation should be in the caption, and the tsne hyperparameters should be in appendix.
 - "we restrict ourselves to... discrete prototypes" not obvious what discrete means here; you're using a continuous embedding space. I assume you mean discrete output tasks (i.e. classification), but this is unclear.
 - It would be nice to see results about computation time; it seems like computing the prototypes might be very intensive.

Questions:
1. Can you contrast your method with prototypical networks? Are prototypes computed in the same way, and then put in a replay buffer along with examples?
2. Why averaging the embeddings? Did you try anything else?
3. How do your task protocols correspond to "single head" and "multi head" evaluation protocols? Why weren't these mentioned? Why did you choose these protocols?
4. The approach of using the same architecture and same memory allowance for all the methods tested seems reasonable to me, but can you think of alternatives, or reasons that this could be unfair to any of the methods?

**Experience Assessment:**

I have published one or two papers in this area.

**Review Assessment: Checking Correctness Of Derivations And Theory:**

N/A

**Review Assessment: Checking Correctness Of Experiments:**

I carefully checked the experiments.

**Review Assessment: Thoroughness In Paper Reading:**

I read the paper thoroughly.

---

### Decision · Program_Chairs · 2019-12-19

**Decision:**

Reject

**Comment:**

The reviewers have provided extensive comments, we encourage the authors to take them into account seriously in further iterations of this work.